# Effect of Treatment with a Combination of Dichloroacetate and Valproic Acid on Adult Glioblastoma Patient-Derived Primary Cells Xenografts on the Chick Embryo Chorioallantoic Membrane

**DOI:** 10.3390/pharmaceutics18010052

**Published:** 2025-12-30

**Authors:** Rūta Skredėnienė, Donatas Stakišaitis, Aidanas Preikšaitis, Angelija Valančiūtė, Vaiva Lesauskaitė, Ingrida Balnytė

**Affiliations:** 1Department of Histology and Embryology, Medical Academy, Lithuanian University of Health Sciences, 44307 Kaunas, Lithuania; ruta.skredeniene@lsmu.lt (R.S.); angelija.valanciute@lsmu.lt (A.V.); ingrida.balnyte@lsmu.lt (I.B.); 2Laboratory of Molecular Oncology, National Cancer Institute, 08660 Vilnius, Lithuania; 3Centre of Neurosurgery, Clinic of Neurology and Neurosurgery, Faculty of Medicine, Vilnius University, 03101 Vilnius, Lithuania; aidanas.preiksaitis@santa.lt; 4Institute of Cardiology, Lithuanian University of Health Sciences, 50009 Kaunas, Lithuania; vaiva.lesauskaite@lsmu.lt

**Keywords:** adult glioblastoma, GFAP, PCNA, p53, EZH2, vimentin, dichloroacetate-valproic acid combination, patient-derived tumor cells, chicken embryo chorioallantoic membrane model

## Abstract

**Background/Objectives:** The ineffectiveness of current treatments for glioblastoma underscores the urgent need for effective alternatives. This study aimed to investigate the effectiveness of sodium dichloroacetate (NaDCA) and a sodium valproate NaDCA combination (NaVPA–NaDCA) on formed patients’ primary cell tumors on the chick embryo chorioallantoic membrane (CAM). **Methods:** Glioblastoma tissue samples were obtained from three patients during tumor surgery. WHO grade IV, IDH wild-type, and a strong positive cytoplasmic GFAP reaction in tumor cells characterized the investigated glioblastoma cases. The tumor cells GBM2-2F, GBM2-3F, and GBM-4M from the patients were examined. Histological examination of tumor invasion into CAM, angiogenesis, and immunohistochemical expression of GFAP-, PCNA-, p53-, EZH2- and vimentin-positive cells were examined. **Results:** No difference in GFAP expression was observed between the patient’s GBM tumor tissue and the tumor formed on CAM from the same patient’s tumor cells. There were no significant differences in invasion or in the frequency of GFAP- and p53-positive cells among the study control groups. The expression of PCNA-, EZH2-, and vimentin-positive cells in control tumors varied significantly. Treatment significantly reduced the incidence of tumor invasion in GBM2-2F and GBM2-4M and did not affect GBM2-3F tumors; treatment also significantly reduced GFAP expression in GBM2-3F and GBM2-4M and did not affect GBM2-2F tumors. The treatment with NaVPA–NaDCA significantly reduced the expression of PCNA, p53, EZH2 and vimentin in the tested tumors. **Conclusions:** Data demonstrated an antitumor effect of NaVPA–NaDCA in an in vivo model of a patient’s primary glioblastoma cells.

## 1. Introduction

Adult glioblastoma (GBM), as WHO Grade IV, IDH wild-type, is classified as primary, the most frequent, highly aggressive, and heterogeneous brain cancer with a poor prognosis, as there is no effective treatment despite multidisciplinary research efforts. The incidence of adult GBM peaks between the ages of 65–74 years, with an average age of 65 years and an overall survival (OS) of 8 months [1,2]. The current treatment regimen consists of tumor maximal surgical resection, adjuvant treatment with temozolomide (TMZ) chemotherapy and radiotherapy [3,4]. TMZ was authorized 20 years ago for the treatment of newly diagnosed GBM [5,6]. The addition of TMZ to radiotherapy prolonged OS by 2.5 months [6], progression-free survival (PFS) is 6 months, and TMZ is ineffective in more than 50% of patients; with standard multimodality therapy, 70% of patients progress within the first year. The GBM 5-year survival rate is less than 5% and is among the lowest of all cancers [6]. It is essential to develop preclinical models for both standard and novel therapies that are as effective and personalized as possible [7].

GBM cells require oxidative phosphorylation (OXPHOS) to survive, and OXPHOS is often a hallmark of cancer stem cells and cells resistant to chemotherapy. Inhibition of OXPHOS function may also effect the tumor microenvironment by reducing hypoxia and improving the anti-tumor immune response, with great potential in GBM in both in vitro and in vivo studies [8,9]. GBM cells exhibit increased expression of pyruvate dehydrogenase kinases (PDK), resulting in enhanced glycolytic activity [10,11,12]. PDK inhibition is a target for GBM treatment [13]. Dichloroacetate (DCA) has a selective effect on cancer cells by inhibiting PDK [14], thereby activating pyruvate dehydrogenase (PDH) [15]. DCA inhibits the growth of U87 MG and PBT24 xenografts on CAM, reduces tumor invasion, and decreases the number of blood vessels in the CAM model, impacting the expression of PCNA and EZH2 in tumor tissue cells [16]. In studies of immunodeficient orthotopic U87 glioblastoma-bearing mice, the combination of DCA and radiotherapy increased animal survival [17].

Valproic acid (VPA) is a histone deacetylase (HDAC) inhibitor that, alone or in combination with other therapies, suppresses glioma growth both in vivo and in vitro [18]. Non-toxic doses of VPA increase the sensitivity of U87 and T98G cells to gefitinib and inhibit cell growth via autophagy [19]. VPA suppresses aerobic glycolysis by reducing the levels of the E2F transcription factor 1, thereby reducing the expression of glycolytic genes, such as glucose-6-phosphate isomerase and phosphoglycerate kinase 1 [20]. VPA may enhance the transport of DCA into the cell through mitochondrial pathways [21,22]. The synergistic action of combining chemotherapy and HDAC inhibitors shows promise for treating GBM and preventing chemotherapy resistance [23]. VPA and DCA are drugs whose pharmacokinetics, blood levels and safety are well known.

It was reported that VPA [24], DCA, the combination of sodium dichloroacetate and sodium valproate (NaVPA–NaDCA), and TMZ have a cell line-dependent effect on the growth, neo-angiogenesis, and expression of proliferating cell nuclear antigen (PCNA), enhancer of zeste homolog 2 (EZH2), and tumor protein p53 (p53) in tumor tissues of adult GBM U87 and T98G cells tumors on CAM. In combination, VPA and DCA act synergistically. The effects in vitro of NaVPA–NaDCA and TMZ on SLC12A2 (Na-K-2Cl cotransporter; NKCC1), SLC12A5 (K-Cl cotransporter; KCC2), SLC5A8 (a Na+-coupled high-affinity transporter for short-chain fatty acids), CDH1 (E-cadherin) and CDH2 (N-cadherin) gene expression in U87 or T98G cells were cell line-dependent, with efficacy data indicating a superior impact of NaVPA–NaDCA compared with TMZ [25].

Inflammation and immune response pathways are directly linked to GBM progression, and their dysregulation is associated with greater treatment resistance and poorer outcomes [26,27]. NaVPA–NaDCA inhibits inflammatory and immune pathways relevant in the genesis of GBM [28,29].

The Chicken Embryo Chorioallantoic Membrane (CAM) model is used to assess approaches to treating GBM, and drug response is evaluated by examining tumor growth, vascularization, invasion, and tumor marker expression via histological analysis [25,30]. The relevance of using patient-derived xenograft (PDX) models for translational oncology research is highlighted in GBM. These models more accurately reflect the genetic and molecular heterogeneity of tumors, which facilitates the development of personalized treatment approaches [7,30,31].

The study aimed to determine differences in the response of GBM IDH-wild-type patients’ derived primary cell xenograft (PDX) on the CAM model to treatment with a NaVPA–NaDCA in terms of tumor growth, invasion, angiogenesis, and the expression of glioma markers such as GFAP, PCNA, p53, EZH2 and vimentin in the tumor tissue on CAM.

GFAP is expressed in human astrocytes, GBM cancer stem cells, and GBM cells and is used for histological identification of GBM [32]. GFAP is a key intermediate filament protein of glial cells, involved in cytoskeleton remodeling and linked to GBM progression, with a functional role in intercellular mitochondrial transfer between GBM cells in response to proapoptotic stimuli [33]. Immunohistochemical analysis of GBM has shown that the GFAP value is associated with worse survival, independent of the methylation status of the O6-methylguanine-DNA methyltransferase (MGMT) promoter [34].

The proliferation of cancer cells correlates with a high degree of tumor malignancy, as assessed by PCNA protein expression [35]. In assessing the association between clinical outcome and GBM immunohistochemical expression and apoptosis in GBM patients, it was found that increased PCNA labelling index and decreased apoptosis index were associated with shorter patient survival [36]. PCNA-associated complexes influence p53 regulators [37].

The tumor marker p53 is a target in the context of developing cancer treatment strategies [38]. There is an interaction between the p53 mutation and wild-type p53; the mutated p53 protein acts as a dominant-negative inhibitor of wild-type p53 tumor suppressor programs and can acquire oncogenic functions [39]. The wild-type, normal p53 protein acts as a tumor suppressor, promoting apoptosis in cancer cells. Mutations in p53 lead to the accumulation of the dysfunctional protein in tumor tissue, while wild-type p53 does not accumulate [40,41]. Drug treatment must target the p53 mutant proteins [40,42,43].

EZH2 is overexpressed in gliomas, correlates with high-grade GBM, and is associated with poorer survival outcomes [44,45]. Its activity contributes to GBM progression by silencing tumor-suppressor genes [46]. The importance of EZH2 inhibition has attracted interest as a therapeutic target [47]. The use of EZH2 inhibitors in GBM models warrants further investigation, as EZH2 inhibition may also have downstream effects on tumor suppressors and other signaling pathways [48].

RNA sequencing data show that vimentin is highly expressed in OXPHOS-resistant cancer cells, particularly in GBM cells. Vimentin plays a vital role in cancer progression, immunotherapy suppression, the development of cancer stem cell properties, and the emergence of drug resistance [49]. The epithelial–mesenchymal transition (EMT) process is characterized by increased vimentin expression in the tumor, which is also associated with loss of cell adhesion [50]. Some studies have reported a link between TMZ resistance and the EMT factor vimentin [51]. Providing new insights into vimentin as a target for GBM treatment strategies [49], considering it as a prospective GBM therapeutic target [52].

The study demonstrates the anticancer effect of NaVPA–NaDCA, reducing PCNA, EZH2, p53, and vimentin expression in GBM IDH-wild-type primary cells GBM2-2F, GBM2-3F, and GBM2-4M tumors on CAM, while the impact on GFAP expression and invasion in CAM has individual effects.

## 2. Materials and Methods

### 2.1. GBM Patient Clinical Data

GBM tissue samples were obtained from three patients who underwent surgery, and their details are provided below. GBM2-2F, 48-year-old female, glioblastoma (WHO Grade IV), IDH wild-type; strong positive cytoplasmic reaction in GFAP tumor cells (surgery 4 October 2021); GBM2-3F, 82-year-old female, glioblastoma (WHO Grade IV), IDH wild-type; strong positive cytoplasmic reaction in GFAP tumor cells (surgery 28 November 2024); GBM2-4M, 46-year-old male; glioblastoma (WHO Grade IV), IDH wild-type; strong positive cytoplasmic reaction in GFAP tumor cells (surgery 1 March 2021). All patients had been diagnosed with GBM for the first time before surgery and had not received any prior anticancer treatment.

The Kaunas Regional Biomedical Research Ethics Committee granted authorization to conduct biomedical research on 8 August 2020, authorization No. BE 2-80. Patients were not given anticancer treatment before surgery. They were operated on at the Neurosurgery Clinic of Vilnius University Santaros Hospital (Santariškių St. 2, Vilnius, Lithuania).

### 2.2. GBM Patient Study Groups

In the GBM tumor groups, tumor growth, the frequency of invasion into the CAM, sub-tumoral changes in the CAM (thickening), PCNA, p53, EZH2, vimentin and GFAP immunohistochemical markers were studied. Cells were treated with 2 mM NaVPA–3 mM NaDCA and the group sample sizes are presented in Table 1.

### 2.3. Investigational Medicinal Preparations

A 2 mM concentration of sodium valproate (NaVPA; Sigma-Aldrich, Steinheim, Germany) and a 3 mM concentration of sodium dichloroacetate (NaDCA; Sigma-Aldrich, Steinheim, Germany) were used in the NaVPA–NaDCA combination. The combination of these therapeutic agents is patented, which recognizes NaVPA–NaDCA as a novel pharmaceutical formulation for treating cancer: National patent—the combination of VPA and NaDCA products is for the treatment of cancer/Valpro rūgšties ir dichloroacetato druskų derinio taikymas vėžio gydymui (Official Bulletin of the State Patent Bureau of the Republic of Lithuania, Patent No. 6874, Int. Cl. A61K 31/00) [53]; European Patent—derivative of valproic acid and dichloroacetate used for the treatment of cancer (Application No. 21168796.7, Publication No. EP3895702, publication date 24 September 2025, Int. Cl. A61K 31/19, A61K 45/06, A61P 35/00 [54]. The doses of NaDCA and VPA were selected based on previously published studies that employed these investigational agents as monotherapies [16,24,55].

### 2.4. The CAM Model

Fresh surgically resected glioblastoma tumor tissue was placed in DMEM medium without Ca^2+^ (Gibco, Grand Island, NY, USA) with 1% 100 IU/mL of penicillin and 100 µg/mL streptomycin (P/S; Gibco, Grand Island, NY, USA). Part of the tissue was fixed in formalin for paraffin embedding. The other part was placed in a Petri dish (Greiner Bio-One, Mosonmagyaróvár, Hungary) and washed repeatedly with Dulbecco’s Phosphate-Buffered Saline (DPBS) without Ca^2+^ and Mg^2+^ (Gibco, Grand Island, NY, USA) to reduce the amount of red blood cells. Any visible connective tissue was removed using sterile tweezers. DMEM with P/S was then added to a Petri dish, and the tissue was minced into small pieces using sterile tweezers within a laminar airflow cabinet under sterile conditions. A cell suspension was acquired by smoothly pipetting a dissected GBM sample in 37 °C DMEM with P/S and 10% fetal bovine serum (FBS; Gibco, Paisley, UK) in a 50 mL conical tube (Thermo Fisher Scientific, Rochester, NY, USA) for 5 min. The time between sample collection and cell suspension acquisition is no more than 20 min.

In compliance with European Union and Lithuanian legal frameworks, the use of the CAM model in research does not necessitate prior approval from an Ethics Committee. Rumšiškės hatchery (Rumšiškės, Lithuania) supplied Cobb-500 freshly fertilized chicken eggs, which were placed in an incubator (Maino Incubators, Oltrona di San Mamette, Como, Italy) for 3 consecutive days at 37 °C under humid conditions (60% relative humidity). Eggs were automatically turned hourly and turning was discontinued on day 3 of embryo development (EDD3). At EDD3, ~2.5 mL of egg white was withdrawn from the blunt end of each egg using a sterile syringe to facilitate detachment of the CAM from the shell. A 1 cm^2^ window was then carefully drilled in the shell and sealed with sterile, transparent film. The eggs were returned to the incubator for GBM patient tumor cell transplantation on CAM at EDD7.

The GBM cell suspension was centrifuged at 800 rpm for 3 min and the supernatant was discarded. The cells in DMEM were resuspended in rat tail collagen, type I (Gibco, New York, NY, USA) (group—control), and in a mixture of collagen, 2 mM NaVPA and 3 mM NaDCA in DMEM (treated group), total volume of preparation for single tumor formation—20 µL. A 20 µL of GBM cell suspension mixture was pipetted onto a 4 mm^3^ piece (1 mm × 2 mm × 2 mm) of a hemostatic, absorbable gelatin sponge (Surgispon, Aegis Lifesciences, Gujarat, India). The gelatin sponge serves as a suitable scaffold for delivering tumor cell suspensions onto the CAM surface, as previously reported by Ribatti [56]. At the EDD7, the tumor-bearing sponge was transplanted on the CAM in proximity to major blood vessels. Structural changes in the tumor were monitored in vivo with a stereomicroscope (SZX2RFA16, Olympus, Tokyo, Japan) from EDD 9 to 12. Tumor images were acquired with a DP92 digital camera connected to CellSens Dimension 1.9 imaging software (Olympus, Tokyo, Japan).

On EDD12, 0.25 mL of fluorescein isothiocyanate–dextran (5 mg/mL; Sigma-Aldrich, St. Louis, MO, USA) was injected into a major CAM blood vessel using an insulin syringe (Chirana T. Injecta, Stara Tura, Slovakia). The tumor and surrounding CAM were harvested and photographed under natural light and ultraviolet illumination using a green fluorescent protein (GFP) filter (Olympus, Tokyo, Japan), as visualized with the aforementioned imaging software (version 1.9). 

### 2.5. Histological and Immunohistochemical Study of the Tumor

Harvested CAM specimens and part of the postoperatively obtained GBM tumor tissue were fixed in 10% neutral and buffered formalin solution for approximately 24 h. Following fixation, samples were embedded in paraffin wax and sectioned at a thickness of 3 µm using a microtome (Leica, Wetzlar, Germany). Hematoxylin and eosin (H-E) and immunohistochemical (IHC) staining methods were performed, as reported [16]. Capturing and visualization of slides stained with H-E and IHC were performed using a microscope (BX 40F4, Olympus, Tokyo, Japan) and a digital camera (XC30, Olympus, Tokyo, Japan), with CellSens Dimension 1.9 software.

Each tumor was evaluated for invasiveness into the CAM using 10× and 40× magnifications in two H-E sections. The invasion was defined as either disruption of the chorionic epithelium (ChE) and/or the presence of tumor cells infiltrating the underlying CAM mesenchyme. In non-invasive cases, the tumor remained localized on the surface of the chorionic epithelium, which retained its structural integrity. Two H-E sections of the CAM region beneath each tumor were captured at 4× magnification for analysis. The average CAM thickness was determined by measuring at 10 distinct points (in µm). All measurements were conducted over a standardized CAM length of 1792 µm.

IHC was performed to detect GFAP, PCNA, p53, EZH2, and vimentin expression. Tissue sections (3 µm thick) were mounted on poly-L-lysine-coated slides (Thermo Fisher Scientific, Branchburg, NJ, USA). Antigen retrieval was performed using the heat-induced epitope retrieval method for 20 min at 95 °C. High pH EnVision FLEX Target Retrieval Solution (pH 9, K8004, Dako, Glostrup, Denmark) was used for GFAP, PCNA, p53, and EZH2, while Low pH EnVision FLEX Target Retrieval Solution (pH 6, K8005, Dako, Glostrup, Denmark) was used for vimentin. Staining was performed using the Shandon CoverPlate System (Thermo Fisher Scientific, Branchburg, NJ, USA). Endogenous peroxidase activity was blocked for 10 min using EnVision FLEX Peroxidase Blocking Reagent (SM801, Dako, Glostrup, Denmark). Primary antibodies were applied at a 1:100 dilution: for PCNA, p53 and EZH2, incubation was conducted for 30 min at room temperature; for GFAP and vimentin, incubation was conducted overnight at 4 °C. The primary antibodies used were: anti-GFAP (EPR1034Y, ab68428, Abcam, Cambridge, UK), anti-PCNA (PC10, Thermo Fisher Scientific, Branchburg, NJ, USA), anti-p53 (aa 211–220, clone PAb240, CBL404), recognizes both mutant and denatured wild-type p53 (Merck, Darmstadt, Germany), anti-KMT6/EZH2 (phospho S21, ab84989, Abcam, Cambridge, UK), and anti-vimentin (V9, M0725, Dako, Glostrup, Denmark). To achieve negative control, the primary antibody was omitted; for positive control, each slide array was used. Detection of immunoreactivity was achieved using a horseradish peroxidase-conjugated polymer dextran system labeled with a mouse antibody (secondary) and a linker (respectively, SM802 and SM804; Dako, Glostrup, Denmark) for 30 min at room temperature. Visualization was completed using 3,3′-diaminobenzidine (DAB, DM827, Dako, Glostrup, Denmark) as the chromogen. For a wash buffer, a Tween 20-containing Tris-buffered saline solution (DM831, Dako, Glostrup, Denmark) was used. Counterstaining was performed with Mayer’s hematoxylin (Sigma-Aldrich, Taufkirchen, Germany). Protein expression of the investigated markers was evaluated by selecting two randomly chosen vision fields (each with a plot area of 23,863.7 µm^2^) within two histological sections of IHC-stained tumor and capturing them at 40× magnification. The marker-positive stained cells and total tumor cells were counted, and the percentage of marker-positive cells was calculated in each histological section.

In Table 2, Table A1, Table A2, Table A3, Table A4 and Table A5, *n* represents the number of histological sections analyzed. For each tumor on CAM, two sections were assessed for histomorphometry measurements (invasion and CAM thickness) and for immunohistochemical evaluation (GFAP, PCNA, p53, EZH2 and vimentin). For each GBM patient tumor tissue sample, a total of 10 sections were analyzed. Histological and histomorphometric analyses were performed by an operator blinded to the experimental groups.

### 2.6. Statistical Analysis

Statistical analysis and visualization were performed using IBM SPSS Statistics version 23.0 (IBM SPSS, Armonk, NY, USA) and GraphPad Prism version 9 (GraphPad Software Inc., San Diego, CA, USA). The frequency of tumor invasion into the CAM was expressed as a percentage (%), and comparisons between study groups were performed using the chi-square test. The Shapiro–Wilk test was applied to assess the normality of data distribution. CAM sub-tumoral thickness and IHC marker expression are presented as medians with corresponding minimum and maximum values. The nonparametric Mann–Whitney U test was used to assess differences between the two independent groups. A *p*-value of <0.05 was considered statistically significant.

## 3. Results

### 3.1. Stereomicroscopic and Histologic Images of Tested Tumors on the CAM In Vivo, the Tumor Ex Ovo, and Their H-E Histological Images

Figure 1 shows stereomicroscopic and histologic images of GBM2-2F, GBM2-3F and GBM2-4M tumors on the CAM.

Figure 1a shows the GBM2-2F biomicroscopy images of tumors on the CAM at the EDD9 and EDD12, the ex ovo tumor with CAM, and the histological H-E images. Comparison of the control tumor EDD9 with EDD12 reveals tumor invasion into the CAM. The H-E image shows damaged chorionic epithelial integrity and tumor invasion into the CAM, with the CAM beneath the tumor appearing thickened and containing multiple vessels. Ex ovo control images show blood vessels around the tumor. Treatment with 2 mM NaVPA–3 mM NaDCA inhibited tumor growth and the formation of a vascular network around the tumor, as evidenced by the absence of a vascular network and no chorionic epithelial damage on H-E images.

Figure 1b shows the biomicroscopy images of the GBM2-3F tumor at EDD9 and EDD12, the excised tumor with the CAM, and the H-E histological images of the tumor on the CAM. GBM2-3F control at EDD9 and 2 mM NaVPA–3 mM NaDCA tumors on CAM have clear boundaries. The control tumor at EDD12 is reduced due to tumor invasion into the CAM mesenchyme. The control ex ovo image shows a pronounced vascular network around the tumor, and the H-E preparation shows compromised chorionic epithelial integrity, tumor invasion, and CAM thickening. Compared to the control, treatment with 2 mM NaVPA–3 mM NaDCA in the H-E image of the tumor reveals CAM thickening beneath the tumor, indicating damage to chorionic epithelial integrity.

Figure 1c shows the biomicroscopy of the GBM2-4M tumor images at EDD9 and EDD12 on the CAM, the tumor ex ovo with the CAM, and the histological H-E images of the tumor in the control and treatment groups.

The GBM2-4M control tumor at EDD9 is visually larger than the tumor at EDD12, as it has penetrated the CAM mesenchyme on day 12, and only part of the tumor is visible on the CAM surface. The histological section of the control H-E tumor shows extensive tumor invasion and is surrounded by CAM mesenchyme. The control ex ovo image shows a vascular network surrounding the tumor. Treatment with 2 mM NaVPA–3 mM NaDCA inhibited tumor invasion (where the tumor is located on the CAM surface) and the formation of a vascular network around the tumor. No lesions of the chorionic epithelium are seen on the H-E image of the treated tumor at EDD12.

### 3.2. GBM2-2F, GBM2-3F and GBM2-4M Tumors’ Growth, Tumor Invasion Rate into CAM, the IMP Impact on Neo-Angiogenesis and CAM Thickness Under the Tumor

Table 2 and Figure 2 present the histomorphometric data on the incidence of tumor invasion into the CAM and the thickness of the CAM beneath the tumor in the studied groups.

There were no significant differences in invasion among the GBM2-2F, GBM2-3F, and GBM2-4M control groups. Compared to matched controls, treatment with 2 mM NaVPA–3 mM NaDCA significantly reduced the incidence of tumor invasion into the CAM in patients GBM2-2F and GBM2-4M. The 2 mM NaVPA–3 mM NaDCA did not affect the invasion of GBM2-3F tumors.

Treatment did not affect the sub-tumor CAM thickness of GBM2-2F and GBM2-3F tumors. Compared to the GBM2-4M control, 2 mM NaVPA–3 mM NaDCA significantly reduced the thickness of the CAM under the tumor. CAM thickness was significantly lower in the GBM2-4M control group than in the GBM2-3F control (*p* = 0.003). Among the treated groups, CAM thickness in GBM2-4M was significantly lower compared to GBM2-2F (*p* = 0.0003) and GBM2-3F (*p* = 0.005).

Figure 3 shows fluorescence stereo-microscopy images after injecting fluorescent dextran into a CAM vessel. Dextran highlighted the vascular network around the tested tumor. Compared to controls, images of 2 mM NaVPA–3 mM NaDCA-treated GBM2-2F, GBM2-3F and GBM2-4M tumors showed blunted neoangiogenesis.

### 3.3. GBM2-2F, GBM2-3F and GBM2-4M Immunohistochemical Examination of GFAP, PCNA, p53, EZH2 and Vimentin Markers Expression in the Tested Tumors on CAM

#### 3.3.1. The GFAP Expression in GBM-Resected Tumor Tissue of Studied Patients and Tumors Formed by Tumor Cells on CAM

Figure 4 and Appendix A, Table A1 present the GFAP expression data in the operated tumor tissue of GBM patients and the GFAP expression in GBM-formed tumors on CAM in control and 2 mM NaVPA–3 mM NaDCA-treated groups.

Histological analysis revealed no differences in GFAP expression between the tumor tissue removed from patients and the control tumor formed by the same tumor cells on CAM (*p* > 0.05). Immunohistochemical analysis showed that the frequency of GFAP-positive cells in the GBM patient tumor tissue and the frequency of GFAP-positive cells in the tumor on the CAM formed from the corresponding patient’s tumor cells (at EDD12) were not significantly different, indicating that the formatted tumors on the CAM are consistent with the histological characteristics of the patient’s tumor.

The frequency of GFAP-positive cells in the control GBM2-2F tumor was 80.24%, in the GBM2-3F tumor, 82.63%, and in the GBM2-4M tumor, 76.00%. Compared with the control, 2 mM NaVPA–3 mM NaDCA significantly reduced the number of GFAP-positive cells in GBM2-3F and GBM2-4M tumors and had no significant effect on GFAP expression in the GBM2-2F tumor on CAM. The frequency of GFAP-positive cells did not differ significantly among the GBM2-2F, GBM2-3F, and GBM2-4M treated groups.

#### 3.3.2. The PCNA Expression of GBM Tumors in Control and Treated Groups

Figure 5 and Appendix A, Table A2 present PCNA expression data from tumors studied on CAM. The frequency of PCNA-positive cells in control tumors—GBM2-2F—88.88%, GBM2-3F—90.95%, and GBM2-4M—was 49.00%.

Compared to the respective control, 2 mM NaVPA–3 mM NaDCA significantly reduced the number of PCNA-positive cells in GBM2-2F, GBM2-3F, and GBM2-4M tumors on CAM. Comparison of control groups showed that GBM2-4M had a significantly lower frequency of PCNA-positive cells than GBM2-2F (*p* = 0.0002) or GBM2-3F (*p* = 0.007). Among the treatment groups, GBM2-2F showed the highest frequency of PCNA-positive cells, when compared with GBM2-3F (*p* = 0.002) and GBM2-4M (*p* < 0.0001). GBM2-4M showed a significantly lower frequency of PCNA-positive cells compared with GBM2-3F (*p* = 0.02).

#### 3.3.3. The p53 Expression of GBM Tumors in Control and Treated Groups

Figure 6 and Appendix A, Table A3 display p53 expression data from the tumors of the GBM tumor-on-CAM model.

The frequency of p53-positive cells in the GBM2-2F control tumor was 50.13%, 62.34% in GBM2-3F, and 56.58% in GBM2-4M. Compared to the control, 2 mM NaVPA–3 mM NaDCA significantly reduced the frequency of p53-positive cells in GBM2-2F, GBM2-3F, and GBM2-4M tumors on CAM. The frequency of p53-positive cells did not differ significantly among GBM2-2F, GBM2-3F, and GBM2-4M in either control or treated groups.

#### 3.3.4. The EZH2 Expression of GBM Tumors in Control and Treated Groups

Figure 7 and Appendix A, Table A4 present the EZH2 expression data for the tumors studied on CAM. The frequency of EZH2-positive cells was 73.48% in the control of GBM2-2F, 91.11% in GBM2-3F, and 58.11% in GBM2-4M.

In the control groups, GBM2-3F had a significantly higher frequency of EZH2-positive cells than GBM2-2F (*p* = 0.01) and GBM2-4M (*p* < 0.0001). In the treatment groups, GBM2-4M had the lowest frequency of EZH2-positive cells, when compared with GBM2-2F (*p* = 0.04) and GBM2-3F (*p* = 0.02). Compared to the control, 2 mM NaVPA–3 mM NaDCA significantly reduced the number of EZH2-positive cells in GBM2-2F, GBM2-3F, and GBM2-4M tumors on CAM.

#### 3.3.5. The Vimentin Expression in the Studied GBM Tumors in the Control and Treated Groups

Figure 8 and Appendix A, Table A5 show the vimentin expression data of the investigated GBM tumors on CAM. The frequency of vimentin-positive cells in the GBM2-2F control tumor was 45.85%, 26.91% in GBM2-3F, and 79.00% in GBM2-4M.

GBM2-4M had a significantly higher frequency of vimentin-positive cells than GBM2-2F (*p* = 0.04) and GBM2-3F (*p* = 0.03) in the control groups. Compared to the control, 2 mM NaVPA–3 mM NaDCA significantly reduced the number of vimentin-positive cells in GBM2-2F, GBM2-3F and GBM2-4M tumors on CAM. GBM2-4M had a significantly higher frequency of vimentin-positive cells than GBM2-2F (*p* = 0.001) and GBM2-3F (*p* < 0.0001) in the treatment groups.

## 4. Discussion

Advances in personalized GBM treatment are closely linked to standard treatment, and new treatment strategies are being explored, including chemotherapy with medicines that target the tumor’s metabolic and molecular properties. Understanding the mechanisms of drug action and drug resistance, while assessing the effect of drugs on changes in cancer prognostic markers, is essential for confirming the effectiveness of GBM treatment [57,58]. The anticancer effect of NaVPA–NaDCA is supported by non-clinical data from in vivo CAM model studies that evaluated the drug’s effects on tumor growth, vascularization, invasion, and tumor tissue marker expression via histological analysis. The data from this study suggest that the CAM model is a valuable tool for evaluating the in vivo efficacy of the investigational drug. The absence of a mature immune response and a capillary network characterizes the CAM. Successful engraftment of transplanted tissue on the CAM requires neovascularization to ensure graft survival. Host capillaries from the chick embryo grow vertically into the implanted GBM tissue [59,60,61,62]. Angiogenesis in the CAM is characterized by increased blood vessel density around the transplanted and invading tumor. Neoplastic cells secrete specific endothelial cell growth factors, cytokines and chemokines that stimulate the growth of recipient blood vessels. Histological changes in CAM located under tumor grafts include increased membrane thickness [59,60,61,63,64,65]. The angiostatic response to NaVPA–NaDCA manifests as a decrease in blood vessel density, which can also be attributed to its proinflammatory and immune-suppressive effects [28,29]. The impact of NaVPA–NaDCA on membrane thickness under the tumors was individual: it did not impact the CAM thickness in GBM2-2F and GBM2-3F tumors. NaVPA–NaDCA significantly reduced the thickness of the CAM of GBM2-4M.

NaVPA–NaDCA, as well as TMZ, in vitro had individual effects on the SLC12A2, SLC12A5, CDH1, CDH2, EZH2 and GFAP gene expressions in human primary cells of GBM (IDH-wild-type tumors), and it was found that NaVPA–NaDCA could have a greater impact, and that may be more effective than TMZ, but the effect is individual [66]. This study investigated the impact of NaVPA–NaDCA on the cancer markers of patient-derived GBM primary cell-formed tumors on CAM under treatment.

GFAP is expressed in human GBM cells and GBM stem cells, serving as a marker for the histological identification of GBM [32]. When comparing the percentage of GFAP-positive cells in the patient’s GBM tumor tissue and in the tumor formed on CAM from the same patient’s tumor cells, IHC evaluation showed no difference. The frequency of GFAP-positive cells did not differ among GBM2-2F, GBM2-3F, and GBM-4M tumor tissues, nor did it differ among the corresponding tumors formed on CAM from the tested patient tumor, demonstrating that GBM control tumors formed from patients’ primary GBM cells on CAM retain tumor immunohistochemical GFAP expression.

The protein GFAP participates in cytoskeletal remodeling, plays a functional role among GBM cell mitochondria in response to a proapoptotic stimulus, and is associated with GBM progression [33]. It was reported that IHC showed that GFAP levels in GBM tumors were linked to worse survival: studies of adult GBM patients showed that the GFAP ≥ 75% had significantly poorer survival, regardless of MGMT promoter methylation status or isocitrate dehydrogenase 1 (IDH1) mutations, leading to the suggestion that quantitative immunohistochemical GFAP assessment could be a new biological marker for predicting survival in GBM patients [34]. Other researchers concluded that there is no strong correlation between GFAP and astrocytoma malignancy, that GFAP-positive populations are highly heterogeneous, and that differentiation between GFAP isoforms GFAPδ and GFAPα could improve the assessment of tumor malignancy status [32].

An increase in GFAP content is a strong indication of reactive astrocyte remodeling. Reactive astrocytes are key regulators of inflammation during glioma genesis, shaping the GBM microenvironment. In the GBM, astrocytes are activated and regulate tumor microenvironment cells through cell–cell contact or the secretion of proinflammatory cytokines, chemokines and other active substances [67]. NaVPA–NaDCA reduces inflammation by inhibiting proinflammatory cytokines and chemokines [28,29], which may counteract the effects of active substances in the GBM microenvironment and alter GFAP protein expression in tumor cells. The 2 mM NaVPA–3 mM NaDCA treatment significantly reduced the number of GFAP-positive cells in GBM2-3F and GBM2-4M tumors on CAM, but did not affect the expression of the GBM2-2F tumor. The Ezh2cKO mouse model shows that Ezh2 depletion in astrocytes significantly increases GFAP expression [68]. Interpreting GFAP-positive cell counts as indicative of astrocyte proliferation, GFAP immunofixation should be evaluated in conjunction with proliferation markers such as PCNA, Ki-67, or other relevant markers [67].

It was reported that GFAP gene expression varies among primary GBM cells, with GBM5-1F and GBM5-3F exhibiting higher expression than GBM5-2F. Treatment with NaVPA–NaDCA significantly upregulated GFAP gene expression in GBM5-2F cells, significantly downregulated it in GBM5-3F cells, and had no effect on gene expression in GBM5-1F cells, while TMZ did not affect GFAP expression in the tested cells [66]. A correlation was observed between shorter survival in patients with grade IV GBM and increased GFAP transcription activity. Similarly, qRT-PCR analysis revealed a trend toward shorter survival with higher GFAP mRNA levels in patients with grade IV astrocytoma. The REMBRANDT dataset also confirmed the association between more prolonged survival in glioma patients and decreased GFAP mRNA expression. Thus, GFAP expression studies may have clinical relevance, as GFAP can be used as a prognostic marker for glioma [34,69].

PCNA studies indicate that increased PCNA expression is associated with malignant tumor characteristics, advanced stage, higher GBM grade, and poor prognosis. Furthermore, alterations in PCNA during tumor treatment may indicate the effectiveness of therapy [35,36,70]. After tumor removal, even supramarginal resection inevitably leaves some tumor cells behind. The remaining GBM cells retain both proliferative and invasive properties, underscoring the importance of effective chemotherapy during tumor progression [71]. Comparison of control groups showed that GBM2-4M has significantly lower PCNA-positive cell expression than GBM2-2F and GBM2-3F. Treatment with 2 mM NaVPA–3 mM NaDCA significantly reduced the number of PCNA-positive cells in all tested tumors on CAM. After treatment with the combination, the frequency of PCNA-positive cells was significantly higher in GBM2-2F tumors than in GBM2-3F and GBM2-4M tumors, and the expression of the marker was significantly lower in GBM2-4M compared to GBM2-3F tumors. It has been reported that the effect of NaVPA–NaDCA on PCNA expression in T98G cells tumors on CAM is synergistic: the frequency of PCNA-positive cells was significantly reduced in those treated with NaVPA–NaDCA, while 3 mM NaDCA did not affect marker expression; the effect of NaVPA–NaDCA and 50 µM TMZ on T98G tumors was comparable [25]. PCNA-associated complexes and p53 activity regulation are closely linked through the p53 signaling pathway [37,72].

p53 mutations cause the accumulation of dysfunctional p53 protein in tumor tissue, while natural p53 is consumed and does not accumulate in tumor tissue [40,41]. The immunochemistry method revealed p53-positive cells in GBM2-2F, GBM2-3F, and GBM2-4M control tumors, with no difference in marker expression among them. The mutated p53 protein can acquire oncogenic functions and act as a dominant inhibitor of the wild-type p53 tumor suppressor and apoptosis pathways in GBM cells [39]. When developing cancer treatment strategies, therapies targeting mutated p53 proteins in tumor tissue should be considered [38,40,42,43]. The 2 mM NaVPA–3 mM NaDCA significantly reduced the frequency of p53-positive cells in GBM2-2F, GBM2-3F and GBM2-4M tumors on CAM, with no difference in marker expression among the treated tumor groups. It was reported that 3 mM NaDCA or 2 mM NaVPA–3 mM NaDCA had no effect on the expression of p53-positive cells in U87 or T98G cell tumors on CAM, while treatment with 50 µM TMZ significantly reduced the number of p53-positive cells in the tumor groups studied, and the expression of p53-positive T98G cells did not differ between the TMZ- and 2 mM NaVPA–3 mM NaDCA-treated groups [25]. The unsatisfactory efficacy of TMZ chemotherapy in treating GBM is associated with high levels of O-6-methylguanine-DNA methyltransferase (MGMT) expression [73]. High doses of TMZ to reduce MGMT activity are limited by toxicity issues [73,74]. The p53 protein is reported to regulate MGMT [75]. Reduced MGMT expression levels in GBM are associated with p53 activation [76,77]. VPA may induce methylation of the MGMT promoter, reduce MGMT expression in GBM cells, and increase the sensitivity of temozolomide-resistant glioma cells and GBM stem cells [78,79]. In vivo, combined treatment with dichloroacetate and radiotherapy improved survival in mice with orthotopic glioblastoma, demonstrating that DCA can increase the sensitivity of GBM cells to radiotherapy by modulating tumor cell metabolism [17].

EZH2 expression in GBM cells correlates with high-grade GBM, which is associated with poorer survival outcomes [44,45]. EZH2 activity is related to the suppression of GBM tumor suppressors and tumor progression [46]. The frequency of EZH2-positive cells in the study’s GBM primary cell control groups ranged from 58.11% to 91.11%. A 2 mM NaVPA–3 mM NaDCA treatment significantly reduced the number of EZH2-positive cells in GBM2-2F, GBM2-3F, and GBM2-4M tumors on CAM. EZH2 inhibition is being investigated as a therapeutic target [47]. Preclinical studies, both in vivo and in vitro, demonstrated that EZH2 inhibitor GSK126 inhibited the viability of primary GBM cells and GBM stem cells, weakened the aggressive malignant phenotype, cell migration and invasion [80]. Interactions between CCl2+ malignant cells and IL-10+ tumor-associated macrophages in the tumor microenvironment, mediated by the EZH2-FOSL2-CCL2 axis, are significant for GBM mesenchymal transition pathways [81]. Studies of T lymphocytes in patients with COVID-19 pneumonia have shown that NaVPA–NaDCA significantly downregulates the expression of the chemokine CCL2 and the IL-10 gene in T cells [28]. The use of EZH2 inhibitors in GBM models warrants further investigation, as EZH2 inhibition may also affect the regulation of tumor suppressors and other signaling pathways [48].

RNA sequencing data indicate that vimentin is highly expressed in OXPHOS-resistant GBM cells and plays a crucial role in tumor progression, immunotherapy suppression, cancer stem cell properties, and the development of drug resistance [49]. Vimentin expression data from the study investigating GBM tumors on CAM showed significant variability: the frequency of vimentin-positive cells was 45.85% in GBM2-2F, 26.91% in GBM2-3F and 79.00% in the GBM2-4M control tumors. The 2 mM NaVPA–3 mM NaDCA significantly reduced the number of vimentin-positive cells in the study tumors on CAM. Vimentin is a key marker of the EMT process, promoting a mesodermal tumor phenotype [82]. An increase in vimentin expression in tumors is associated with EMT, characterized by a loss of cell adhesion [50]. Some studies have reported a link between TMZ resistance and vimentin expression [51]. The EMT process not only causes drug resistance in GBM but also facilitates tumor invasion [83]. GBM can hijack brain communication for tumor invasion through tumor microtubule formation, and vimentin plays an essential role in this process. In clinical GBM patients, immunofluorescence staining for GFAP and vimentin showed a progressive increase in vimentin expression, correlating with higher tumor grade. Vimentin-positive cells were partially colocalized with GFAP-positive cells, suggesting that GFAP-positive, vimentin-positive cells are associated with GBM progression and may influence tumor invasion [84]. Vimentin expression is inversely correlated with the prognosis of glioma patients [84]. Vimentin is considered a promising therapeutic target for GBM [49,52].

It would also be essential to discuss the advantages of the CAM model compared to conventional PDX models that involve implanting patient tumor cells or tumor fragments into immunodeficient mice. They are the mainstay of preclinical and clinical drug evaluation, as they preserve key histopathological and genetic features of the primary tumor better than cell lines or genetically modified mice [31]. A significant research hurdle is the limited use of PDXs in rodents, as patient-derived tumor cells often fail to grow in these animals [7]. Without a technically complex orthotopic model of brain tumors, the adherence of human glioma fragments in mice is very low. Repeated tissue transfers into mice can replace the human tumor microenvironment with mouse components, potentially altering drug response [85,86]. Notably, there is a discrepancy in vimentin expression between patient samples and tumor xenografts in mice [84]. The successful engraftment rate for GBM on CAM was 98.3% [30], and in heterotopic mouse PDX glioma models, success rates ranged from 38% to 69% [87,88]. The orthotopic brain tumor model had a success rate of 76–90% [89,90]; however, a long-standing problem in developing GBM PDX models is the almost complete failure to engraft IDH1-mutant samples when transplanted into mice [87,91]. In contrast, PDX-CAM is practically always successful regardless of glioma IDH status [30,59]. PDX-CAM results are available within a week, whereas PDXs in rodents take a few times longer.

The synergistic action of chemotherapy and the HDAC inhibitor VPA is promising for treating GBM and preventing chemotherapy resistance. [23,92]. The novelty of the study lies in the finding that VPA may enhance DCA transport into the cell via demethylated SLC5A8, the DCA transporter [21,22]. Such synergism may be relevant because it could allow a reduction in the DCA dose and the risk of its undesirable effects.

The limitation of the study is that only three patients with GBM primary cell tumors on CAM were studied. However, studies have shown that the NaVPA–NaDCA combination is effective in inhibiting GBM carcinogenesis. Another important aspect is that further studies on GBM primary cells are needed to determine the effect of the combination on immune and inflammatory pathways important for GBM progression.

## 5. Conclusions

Glioblastoma tumors formed from patients’ primary cells on CAM retain tumor GFAP expression, indicating that the CAM model is reliable for studying the properties of the tumor.Preclinical in vivo studies using the CAM model indicate that NaVPA–NaDCA has an anticancer effect on GFAP, p53, PCNA, EZH2, and vimentin GBM markers expression, which is important for carcinogenesis.The CAM model is informative for assessing the efficacy of the investigational drug in glioma primary cell tumors and may be a relevant tool for evaluating the potential effectiveness of chemotherapy; however, the expected effect of the drug should be assessed individually.

## 6. Patents

National patent—the combination of NaVPA and NaDCA products is for the treatment of cancer/Valpro rūgšties ir dichloroacetato druskų derinio taikymas vėžio gydymui (Official Bulletin of the State Patent Bureau of the Republic of Lithuania, Patent No. 6874, Int. Cl. A61K 31/00) [53]; European Patent—derivative of valproic acid and dichloroacetate used for the treatment of cancer (Application No. 21168796.7, Publication No. EP3895702, publication date 24 September 2025, Int. Cl. A61K 31/19, A61K 45/06, A61P 35/00 [54].

## Figures and Tables

**Figure 1 pharmaceutics-18-00052-f001:**
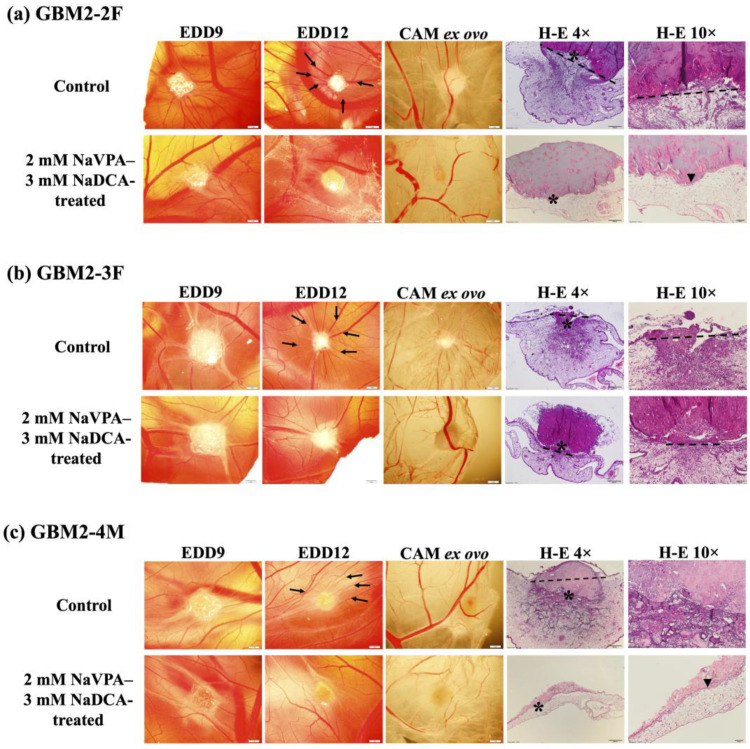
Stereomicroscopic and histologic images of GBM2-2F (**a**), GBM2-3F (**b**) and GBM2-4M (**c**) tumors in vivo on the CAM at EDD9, EDD12, ex ovo at EDD12 (scale bar—1 mm), and H-E stained histologically (4× scale bar—200 µm, 10×—50 µm). Arrows indicate the “spoked wheel”; dotted line –disruption of chorionic epithelium; asterisk—the enlarged region in H-E 10× pictures; arrowhead—intact chorionic epithelium.

**Figure 2 pharmaceutics-18-00052-f002:**
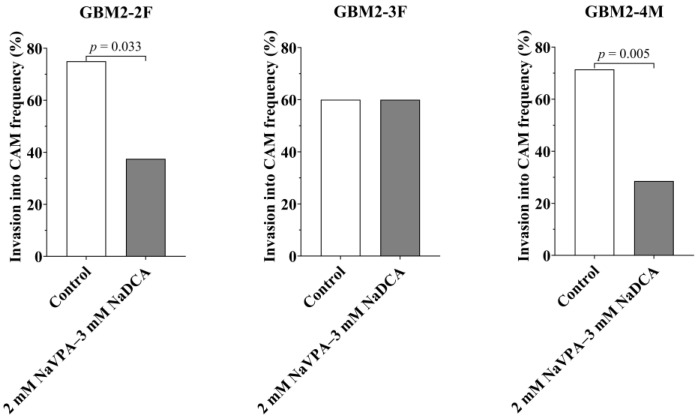
The frequency of tumor invasion into CAM in the study tumor in the CAM groups: GBM2-2F, GBM2-3F, and GBM2-4M.

**Figure 3 pharmaceutics-18-00052-f003:**
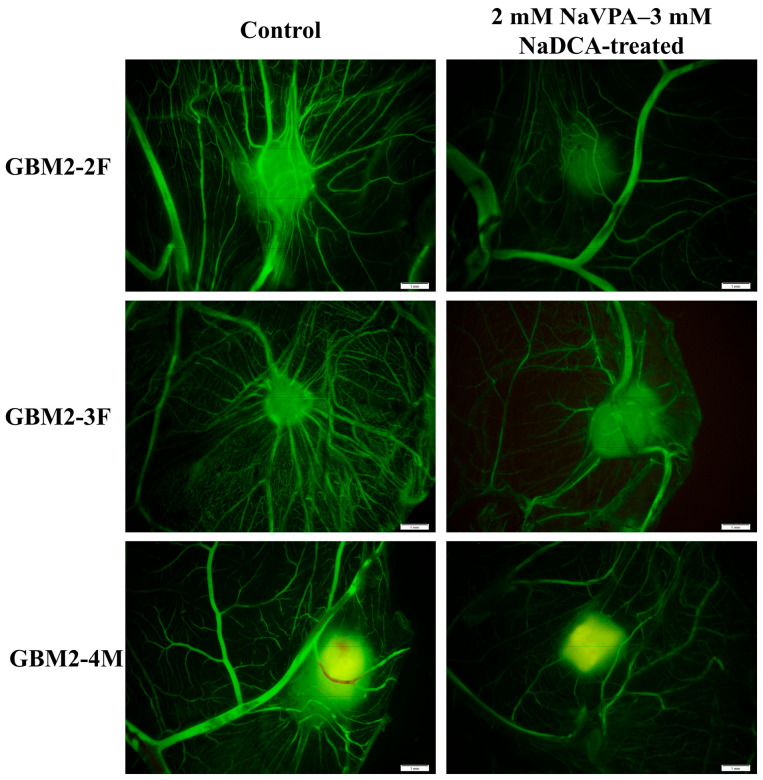
Fluorescent stereomicroscopic images of GBM (glioblastoma) patient tumors (fluorescent dextran highlights the tumor and surrounding blood vessels). Scale bar—1 mm.

**Figure 4 pharmaceutics-18-00052-f004:**
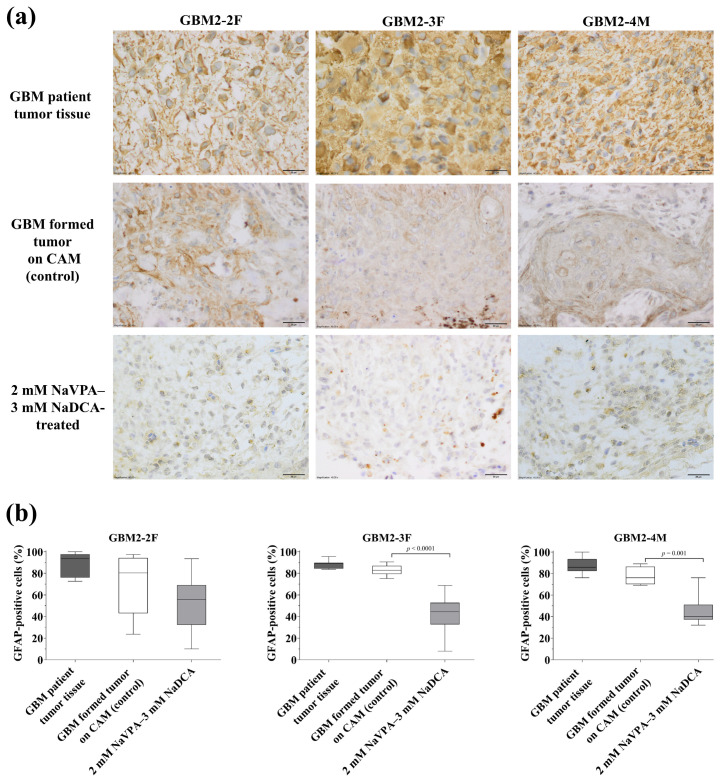
Immunohistochemical GFAP expression in the GBM patient tumor tissue and the GBM tumors on CAM formed from the corresponding patient’s tumor cells in control and 2 mM NaVPA–3 mM NaDCA-treated groups. (**a**) Brown cytoplasm shows GFAP-positive cells. Scale bar—20 µm; (**b**) The percentage of GFAP-positive cells.

**Figure 5 pharmaceutics-18-00052-f005:**
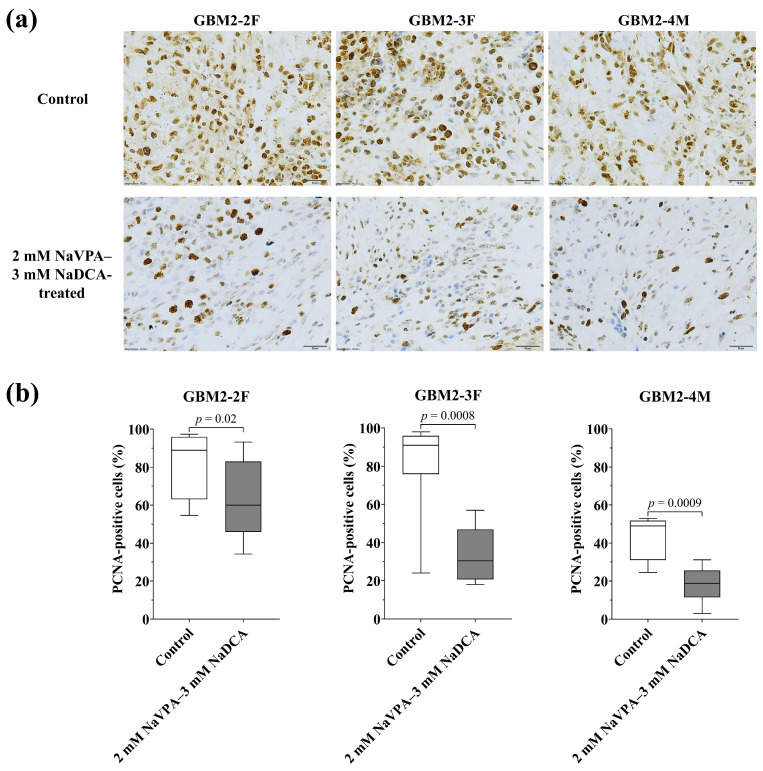
Immunohistochemical PCNA expression in the GBM tumors on CAM. (**a**) Brown nucleus shows PCNA-positive cells in control and 2 mM NaVPA–3 mM NaDCA-treated tumors on CAM. Scale—20 µm; (**b**) The percentage of PCNA-positive cells.

**Figure 6 pharmaceutics-18-00052-f006:**
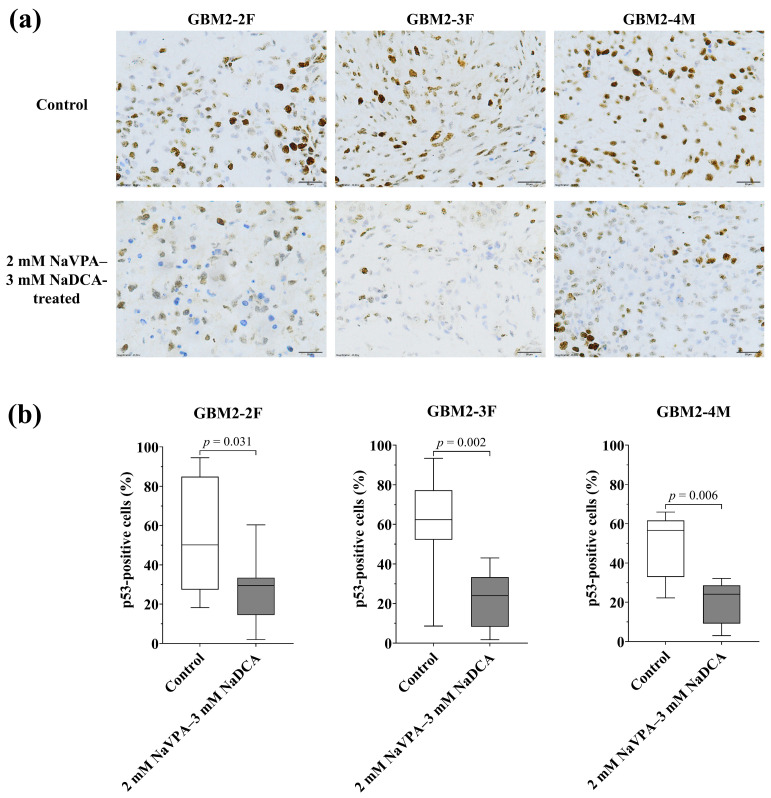
Immunohistochemical p53 expression in the GBM tumors on CAM. (**a**) Brown nucleus shows p53-positive cells in control and 2 mM NaVPA–3 mM NaDCA-treated tumors on CAM. Scale—20 µm; (**b**) The percentage of p53-positive cells.

**Figure 7 pharmaceutics-18-00052-f007:**
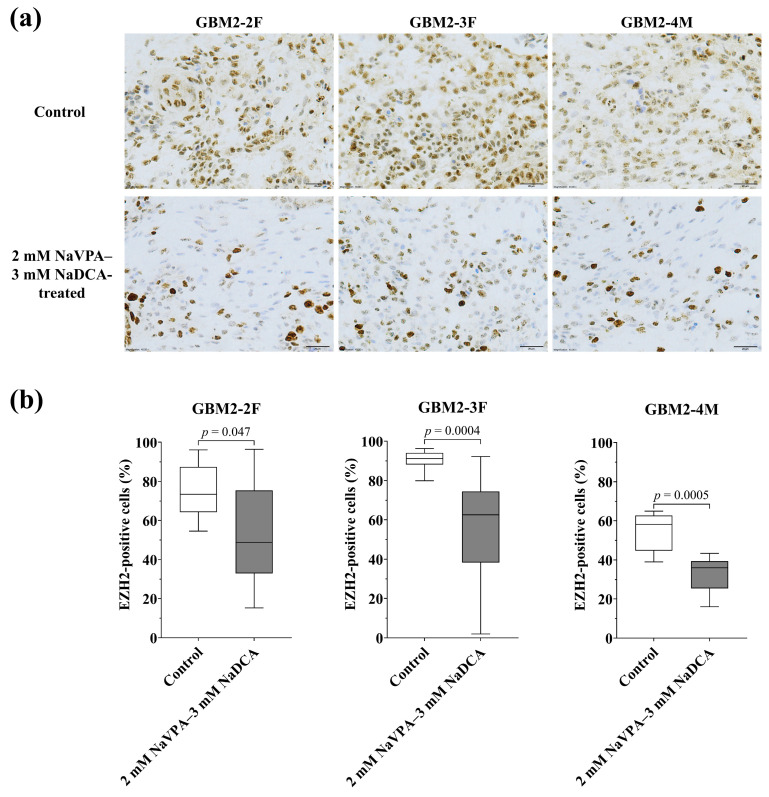
Immunohistochemical EZH2 expression in the GBM tumors on CAM. (**a**) Brown nucleus shows EZH2-positive cells in control and 2 mM NaVPA–3 mM NaDCA-treated tumors on CAM. Scale—20 µm; (**b**) The percentage of EZH2-positive cells.

**Figure 8 pharmaceutics-18-00052-f008:**
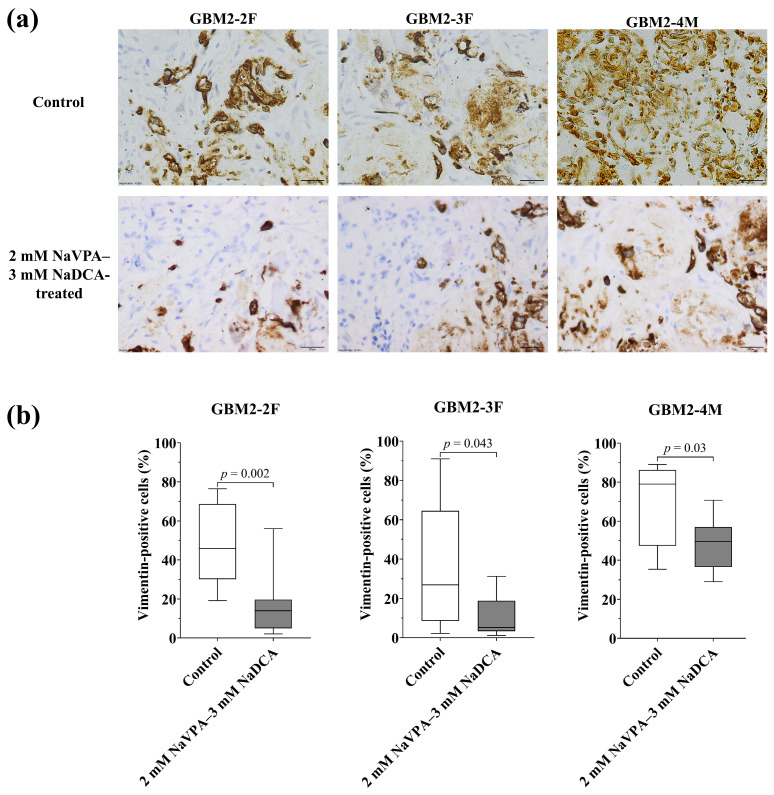
Immunohistochemical vimentin expression in the GBM tumors on CAM. (**a**) Brown cytoplasm shows vimentin-positive cells in control and 2 mM NaVPA–3 mM NaDCA-treated tumors on CAM. Scale—20 µm; (**b**) The percentage of vimentin-positive cells.

**Table 1 pharmaceutics-18-00052-t001:** Sample size of GBM tumors on CAM and tumor marker expression in the studied groups.

Patient Study Group	Invasion, CAM Thickness			GFAP, PCNA, p53, EZH2, Vimentin		
	Number of Tumors on CAM					
	GBM2 -2F	GBM2 -3F	GBM2 -4M	GBM2 -2F	GBM2-3F	GBM2 -4M
Control	8	10	7	4	6	4
2 mM NaVPA–3 mM NaDCA-treated	8	10	7	6	5	5

**Table 2 pharmaceutics-18-00052-t002:** The invasion into CAM frequency and the CAM thickness data of the tested tumors.

Patient Study Group	*n*	Invasion Frequency (%)	CAM Thickness (µm), Median (Range)
GBM2-2F			
Control	16	75.00	201.80 (47.32–677.70)
2 mM NaVPA–3 mM NaDCA-treated	16	37.50 ^a^	403.10 (150.30–672.20)
GBM2-3F			
Control	20	60.00	398.6 0 (98.00–675.60)
2 mM NaVPA–3 mM NaDCA-treated	20	60.00	422.70 (100.05–782.40)
GBM2-4M			
Control	14	71.43	201.30 (44.77–350.10)
2 mM NaVPA–3 mM NaDCA-treated	14	28.57 ^b^	171.40 (120.10–356.50) ^c^

^a^ *p* = 0.033, compared with control; ^b^ *p* = 0.005, compared with control; ^c^ *p* = 0.037, compared with control.

## Data Availability

Data will be made available upon request.

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
