# Peer review of "Effect of Treatment with a Combination of Dichloroacetate and Valproic Acid on Adult Glioblastoma Patient-Derived Primary Cells Xenografts on the Chick Embryo Chorioallantoic Membrane"

_pharmaceutics, 2025, doi:10.3390/pharmaceutics18010052_

Round 1

Reviewer 1 Report

Comments and Suggestions for Authors

The submitted manuscript evaluates the effects of dichloroacetate (DCA) and valproic acid (VPA) on patient-derived primary cultures of Grade 4 glioblastoma using a chick chorioallantoic membrane (CAM) xenograft model. The study explores the potential of combining both agents as a therapeutic approach. The manuscript is written clearly, and the presented data support the main conclusions.

However, several points require clarification before the manuscript can be considered for publication.

1. Section 2.4: The description of primary cell acquisition should be more detailed, as several methodological elements are missing. The authors should provide additional information on the procedures used for cell extraction, including whether any steps such as debris removal or red blood cell lysis were performed. They should also describe how cell quality was assessed (e.g., viability testing) and report the number of cells obtained. In addition, please specify how many cells were used for each tumor inoculation in the CAM model.

2. Experimental groups (lines 204–207): The description of the experimental groups is not sufficiently clear and needs correction. Please clearly specify how the cells used for grafting were prepared and what additives or solutions were applied in each experimental group, including the control group.

3. Discussion section: The authors provide a detailed interpretation of their findings and relate them well to the current literature. However, the section lacks a broader synthesis of the results and their translational relevance. The authors are encouraged to more clearly highlight the novelty of their findings, the potential clinical implications, and possible future research directions. For example, they may discuss the value of expanding the patient sample set in future studies, comparing the proposed agents with standard chemotherapies for glioblastoma, or examining molecular mechanisms such as therapy-induced chemoresistance?

Minor remarks:
Stereoscopic and histologic images (Figures 1, 2, and 3) could be combined into a single figure and presented at a larger size to allow clearer visualization of neoangiogenesis and invasion features, as well as easier comparison of these characteristics between the patients.

Line 150: the data description is incomplete.

After addressing these points, the manuscript will be more methodologically transparent and provide a stronger basis for further translational and preclinical work.

Author Response

For research article

Response to Reviewer 1 Comments

1. Summary

Thank you very much for taking the time to review this manuscript. Please find the detailed responses below and the corresponding revisions/corrections in track changes in the re-submitted files.

2. Questions for General Evaluation

Reviewer’s Evaluation

Response and Revisions

Does the introduction provide sufficient background and include all relevant references?

Yes

Are all the cited references relevant to the research?

Yes

Is the research design appropriate?

Can be improved

Are the methods adequately described?

Yes

Are the results clearly presented?

Yes

Are the conclusions supported by the results?

Can be improved

3. Point-by-point response to Comments and Suggestions for Authors

Comments 1:

Section 2.4: The description of primary cell acquisition should be more detailed, as several methodological elements are missing. The authors should provide additional information on the procedures used for cell extraction, including whether any steps such as debris removal or red blood cell lysis were performed. They should also describe how cell quality was assessed (e.g., viability testing) and report the number of cells obtained. In addition, please specify how many cells were used for each tumor inoculation in the CAM model.

Response 1:

Thank you for your important comment. The article has been updated accordingly in section 2.4 (L188-L192). The removal of debris is described in the Methods.

We reduced the number of red blood cells by repeatedly washing the tumor tissue, which had been shredded up in the medium. We did not apply erythrocyte lysis to avoid any undesirable effects on the tumor cells, so that their properties would not be altered and the environment would remain as natural as possible. The tumor microenvironment (TME) of glioblastoma (GBM) is a highly complex, heterogeneous, dynamic system consisting of cancer cells, various brain and immune cells, and 50 percent or more of the tumor cells are made up of other cells in the tumor environment. The TME is influenced by cell interactions and changes in immune cell activity. The interactions among GBM cells and components, and their relationship to carcinogenesis, are important areas of research in the search for therapeutic targets (Sharma et al. 2023). We did not attempt to isolate pure GBM cells because our method maintained an environment closer to that of the patient's tumor.

We did not determine cell viability. Preparation of the suspension took up to 20 minutes, and preparation of the tumor content took the same amount of time (its preparation is described in the Methods section). Tumor growth on CAM and histological and histochemical GFPA expression studies indicate that tumor cells remained viable. Other authors indicate that the processing of patient tumor tissue should not exceed 2–3 hours. (Kim SS et al., 2015; Binder ZA et al. 2016; Rose M et al. 20221).

We did not evaluate cell viability because the entire sample was used immediately after processing, as the aim of the study was to preserve the complete heterogeneity of the primary cells and avoid the loss of fragile primary cell properties during additional processing steps. In addition, to avoid loss of viability, the cells were implanted immediately after the implantable tumor content formed.

We did not count the cells because the suspension contained not only individual cells but also very small pieces of tumor tissue. Our experience shows that tumor studies can also be performed by transplanting tumor tissue onto CAM (Uloza V et al. 2015; 2021; 2021; Balčiūnienė N et al. 2009). Our measure was the uniform-suspension-volume transfer on CAM.

References

1.     Sharma P, Aaroe A, Liang J, Puduvalli VK. Tumor microenvironment in glioblastoma: Current and emerging concepts. Neurooncol Adv. 2023 Feb 23;5(1):vdad009. doi: 10.1093/noajnl/vdad009.

2.     Kim SS, Pirollo KF, Chang EH. Isolation and Culturing of Glioma Cancer Stem Cells. Curr Protoc Cell Biol. 2015 Jun 1;67:23.10.1-23.10.10. doi: 10.1002/0471143030.cb2310s67.

3.     Binder ZA, Wilson KM, Salmasi V, Orr BA, Eberhart CG, Siu IM, Lim M, Weingart JD, Quinones-Hinojosa A, Bettegowda C, Kassam AB, Olivi A, Brem H, Riggins GJ, Gallia GL. Establishment and Biological Characterization of a Panel of Glioblastoma Multiforme (GBM) and GBM Variant Oncosphere Cell Lines. PLoS One. 2016 Mar 30;11(3):e0150271. doi: 10.1371/journal.pone.0150271.

4.     Rose M, Cardon T, Aboulouard S, Hajjaji N, Kobeissy F, Duhamel M, Fournier I, Salzet M. Surfaceome Proteomic of Glioblastoma Revealed Potential Targets for Immunotherapy. Front Immunol. 2021 Sep 27;12:746168. doi: 10.3389/fimmu.2021.746168.

5.     Uloza V, Kuzminienė A, Palubinskienė J, Balnytė I, Ulozienė I, Valančiūtė A. Laryngeal carcinoma experimental model suggests the possibility of tumor seeding to gastrostomy site. Med Hypotheses. 2021 May;150:110573. doi: 10.1016/j.mehy.2021.110573.

6.     Uloza V, Kuzminienė A, Palubinskienė J, Balnytė I, Ulozienė I, Valančiūtė A. An Experimental Model of Human Recurrent Respiratory Papillomatosis: A Bridge to Clinical Insights. Laryngoscope. 2021 Mar;131(3):E914-E920. doi: 10.1002/lary.29093.

7.     Uloza V, Kuzminienė A, Šalomskaitė-Davalgienė S, Palubinskienė J, Balnytė I, Ulozienė I, Šaferis V, Valančiūtė A. Effect of Laryngeal Squamous Cell Carcinoma Tissue Implantation on the Chick Embryo Chorioallantoic Membrane: Morphometric Measurements and Vascularity. Biomed Res Int. 2015;2015:629754. doi: 10.1155/2015/629754.

8.     Balciūniene N, Tamasauskas A, Valanciūte A, Deltuva V, Vaitiekaitis G, Gudinaviciene I, Weis J, von Keyserlingk DG. Histology of human glioblastoma transplanted on chicken chorioallantoic membrane. Medicina (Kaunas). 2009;45(2):123-31.

Comments 2:

Experimental groups (lines 204–207): The description of the experimental groups is not sufficiently clear and needs correction. Please clearly specify how the cells used for grafting were prepared and what additives or solutions were applied in each experimental group, including the control group.

Response 2: Thank you for your comment. We have corrected the description of the experimental groups in accordance with your comment (L210-L214). The tumor volume was 20 μl and consisted of GBM cells and type I collagen from rat tails (control group). The treated groups also included 2 mM NaVPA and 3 mM NaDCA (L212-L213).

Comments 3:

Discussion section: The authors provide a detailed interpretation of their findings and relate them well to the current literature. However, the section lacks a broader synthesis of the results and their translational relevance. The authors are encouraged to more clearly highlight the novelty of their findings, the potential clinical implications, and possible future research directions. For example, they may discuss the value of expanding the patient sample set in future studies, comparing the proposed agents with standard chemotherapies for glioblastoma, or examining molecular mechanisms such as therapy-induced chemoresistance?

Response 3.

Thank you for your comment. In the discussion section, we added the novelty of our study, i.e., the mechanism of synergy between VPA and NaDCA, which allows us to predict a reduction in NaDCA dose and thus reduce the risk of drug-related adverse effects. We believe that we have partially addressed the other aspects of your comment by revising the conclusions.

Minor remarks:
Stereoscopic and histologic images (Figures 1, 2, and 3) could be combined into a single figure and presented at a larger size to allow more precise visualization of neoangiogenesis and invasion features, as well as easier comparison of these characteristics between the patients.

Response: We have combined Fig. 1,2 and 3 into a single figure.

Line 150: the data description is incomplete.

Response: Thank you for your important comment. A sentence has been added to the patient description: All patients had been diagnosed with GBM for the first time before surgery and had not received any prior anticancer treatment (L159-161).

5. Additional clarifications

Changes, additions, corrections, and revision of the English in the manuscript were made using Track Changer. Therefore, all changes are visible: changed figure numbering, text transfers, and text corrections and additions.

We are grateful to the Reviewer. The comments were very valuable, and we hope that the corrections made have improved the manuscript.

Sincerely,

Donatas Stakišaitis

Reviewer 2 Report

Comments and Suggestions for Authors

The main purpose of the present study was to evaluate a combination of Dichloroacetate and Valproic Acid on xenografts obtained on the chorioallantoic membrane of chicken embryos.It is an important study with a major impact on oncology, but certain adjustments are necessary for the study to be adequate and reproducible.

  1. How did you select the doses used for the study? Was the patent obtained by the group of researchers who also contributed to this study?
  2. Line 169. Could the authors clarify why additional concentrations were not tested to assess the dose–response relationship and antiproliferative potential of the treatment? Testing a wider concentration range (and reporting IC₅₀/EC₅₀ values where appropriate) would strengthen conclusions regarding efficacy.

    Line 192. Please confirm the timeline between tumour excision and tissue processing. Do I understand correctly that 20 minutes elapsed from excision to the start of processing? If so, please state this explicitly and justify that this interval did not affect tissue viability or downstream measurements.

    Lines 200–202. Please explain why the interval between egg preparation and inoculation with the glioblastoma cell suspension was necessary. Provide a clear description of any manipulations performed during this interval and justify their duration.

    Line 206. For transparency and reproducibility, please report the exact concentrations used in the mixture of test substances applied to the model.

    Line 207. Please state whether cell viability was assessed after preparation of the suspension and, if so, which assay was used and the quantitative results. Additionally, estimate how many cells were present in the 20 µL of suspension applied to the gelatin sponge, and describe how that estimate was obtained.

    Line 212. Please report the number of embryonated eggs used per experimental group (per sample), and indicate whether this number was determined by a power calculation or other rationale.

    Line 213. Was the stated incubation period sufficient for tumour formation on the CAM? Please provide evidence (references or pilot data) supporting the chosen incubation duration for reliable tumour establishment.

    Line 217. Please indicate the gauge/size of the needle used for FITC inoculation, and provide a brief justification for this choice.

    Line 182. The protocol description is unclear regarding how the mixture of test substances was inoculated onto the CAM. Please provide a step-by-step account of how the tested mixture was prepared and delivered to the model (timing relative to cell inoculation, vehicle used, physical placement on the CAM, etc.).

    Line 335. Please describe in detail the methods used to obtain the histomorphometric data (e.g., staining procedures, image acquisition, software, thresholds, number of fields per sample, operator blinding). These methods should appear in the Materials and Methods section, as the current description is insufficient for reproducibility.

    Figures. Please add explanatory legends beneath each figure that include the sample size (n), summary statistics shown, statistical tests used, exact p-values (or ranges), and definition of error bars. This information should also be referenced in the main text when interpreting the figures.

    Line 361. Please specify the time interval after inoculation of the cell suspension at which neovascularization was first observed and how it was assessed (macroscopically, microscopically, or via imaging).

    The quality of the histological images should be improved for publication. Please provide higher-resolution micrographs, indicate magnification and scale bars, and revise the image descriptions to reflect these changes.

    Please ensure consistent and correct formatting of the terms in vivo and in vitro throughout the manuscript (italicized).

  3. To strengthen the rigor and reproducibility of the study, several methodological improvements are recommended. First, a dose–response experiment should be conducted using multiple concentrations, reporting IC₅₀/EC₅₀ values where applicable, and increasing the number of biological replicates to enhance statistical power. Second, cell viability and accurate cell counts should be reported. For histomorphometric analyses, full methodological details should be provided, including microscope model and objective magnification, image acquisition and analysis software, analysis parameters, number of sections and fields evaluated per sample, and checks for inter-observer variability.
  4. The limitations of the study are discussed and refer only to the small number of samples used to evaluate the inhibitory potential of the NaVPA–NaDCA combination. My question is whether the results obtained in the CAM model can be extrapolated to animal models with xenograft transplants?
  5. I consider it important to reassess the conclusions, highlighting the novel aspects revealed by this study. Additionally, any limitations and implications for future research should be clearly addressed to contextualize the findings.

Author Response

For research article

Response to Reviewer 2 Comments

1. Summary

Thank you very much for taking the time to review this manuscript. Please find the detailed responses below and the corresponding revisions/corrections in track changes in the re-submitted files.

2. Questions for General Evaluation

Reviewer’s Evaluation

Response and Revisions

Does the introduction provide sufficient background and include all relevant references?

Yes

Are all the cited references relevant to the research?

Must be improved

Is the research design appropriate?

Must be improved

Are the methods adequately described?

Must be improved

Are the results clearly presented?

Must be improved

Are the conclusions supported by the results?

Must be improved

3. Point-by-point response to Comments and Suggestions for Authors

Comments 1: How did you select the doses used for the study? Was the patent obtained by the group of researchers who also contributed to this study?

Response: The doses of NaDCA and NaVPA were selected based on prior studies and published data, in which the test compounds were used as monotherapy or in combination, as indicated in L183 (1–5).

We supplemented the article with information on the study's novelty, noting the synergistic effect of VPA and DCA, which allows for a lower dose of NaDCA than when NaDCA is used as monotherapy.

The patent's inventors are a group of researchers, some of whom are also authors of this article.

References

1.     Stakišaitis D, Damanskienė E, Curkūnavičiūtė R, Juknevičienė M, Alonso MM, Valančiūtė A, Ročka S, Balnytė I. The Effectiveness of Dichloroacetate on Human Glioblastoma Xenograft Growth Depends on Na+ and Mg2+ Cations. Dose Response. 2021 Feb 27;19(1):1559325821990166. doi: 10.1177/1559325821990166.

2.     Skredėnienė R, Stakišaitis D, Valančiūtė A, Balnytė I. In Vivo and In Vitro Experimental Study Comparing the Effect of a Combination of Sodium Dichloroacetate and Valproic Acid with That of Temozolomide on Adult Glioblastoma. Int J Mol Sci. 2025 Jul 15;26(14):6784. doi: 10.3390/ijms26146784.

3.     Gečys D, Akramas L, Preikšaitis A, Balnytė I, Vaitkevičius A, Šimienė J, Stakišaitis D. Comparison of the Effect of the Combination of Sodium Valproate and Sodium Dichloroacetate on the Expression of SLC12A2SLC12A5CDH1CDH2EZH2, and GFAP in Primary Female Glioblastoma Cells with That of Temozolomide. Pharmaceutics. 2025 Sep 4;17(9):1161. doi: 10.3390/pharmaceutics17091161.

4.     Kavaliauskaitė D, Stakišaitis D, Martinkutė J, Šlekienė L, Kazlauskas A, Balnytė I, Lesauskaitė V, Valančiūtė A. The Effect of Sodium Valproate on the Glioblastoma U87 Cell Line Tumor Development on the Chicken Embryo Chorioallantoic Membrane and on EZH2 and p53 Expression. Biomed Res Int. 2017;2017:6326053. doi: 10.1155/2017/6326053.

5.     Šlekienė L, Stakišaitis D, Balnytė I, Valančiūtė A. Sodium Valproate Inhibits Small Cell Lung Cancer Tumor Growth on the Chicken Embryo Chorioallantoic Membrane and Reduces the p53 and EZH2 Expression. Dose Response. 2018 Apr 26;16(2):1559325818772486. doi: 10.1177/1559325818772486.

Comments 2:

Line 169. Could the authors clarify why additional concentrations were not tested to assess the dose–response relationship and antiproliferative potential of the treatment? Testing a wider concentration range (and reporting IC₅₀/EC₅₀ values where appropriate) would strengthen conclusions regarding efficacy.
Response: Thank you for the comment. Only one concentration of the combination was tested in this study because the amount of patient material was limited; increasing the number of doses would reduce the number of replicates, thereby weakening the statistical power. Our study aimed to determine whether NaVPA–NaDCA affects patient GBM cells in the CAM model and to compare the individual effects. The aim of our study was not to investigate pharmacological parameters IC₅₀ or EC₅₀ (as mentioned above, this is limited by the amount of tumor material available after surgery). The CAM model is not suitable for obtaining pharmacological IC₅₀ or EC₅₀ data; we aimed to determine the effects of the applied combination doses on tumor growth, invasion into the CAM, angiogenesis, and the expression of the studied markers in the tumor, and to compare the effect of investigational medicine on tumors from different patients. IC₅₀ is useful for inhibitors, indicating the concentration required to block activity. It should be noted that the effects of NaDCA, NaVPA, or the combination are individual, depend on the cell line, and it is difficult to perform such studies using the CAM model in practice. This indicator is more applicable to studies using animal models.

Line 192. Please confirm the timeline between tumour excision and tissue processing. Do I understand correctly that 20 minutes elapsed from excision to the start of processing? If so, please state this explicitly and justify that this interval did not affect tissue viability or downstream measurements.

Response: In response to this comment, we would like to clarify that approximately 20 minutes elapsed between the receipt of the tumor tissue after surgery and the receipt of the cell suspension. The duration of tumor formation before implantation on CAM also did not exceed 20 minutes. Other authors reported that the processing of patient tumor tissue should not exceed 2–3 hours (6–8).

References

6.     Kim SS, Pirollo KF, Chang EH. Isolation and Culturing of Glioma Cancer Stem Cells. Curr Protoc Cell Biol. 2015 Jun 1;67:23.10.1-23.10.10. doi: 10.1002/0471143030.cb2310s67.

7.     Binder ZA, Wilson KM, Salmasi V, Orr BA, Eberhart CG, Siu IM, Lim M, Weingart JD, Quinones-Hinojosa A, Bettegowda C, Kassam AB, Olivi A, Brem H, Riggins GJ, Gallia GL. Establishment and Biological Characterization of a Panel of Glioblastoma Multiforme (GBM) and GBM Variant Oncosphere Cell Lines. PLoS One. 2016 Mar 30;11(3):e0150271. doi: 10.1371/journal.pone.0150271.

8.     Rose M, Cardon T, Aboulouard S, Hajjaji N, Kobeissy F, Duhamel M, Fournier I, Salzet M. Surfaceome Proteomic of Glioblastoma Revealed Potential Targets for Immunotherapy. Front Immunol. 2021 Sep 27;12:746168. doi: 10.3389/fimmu.2021.746168.

Lines 200–202. Please explain why the interval between egg preparation and inoculation with the glioblastoma cell suspension was necessary. Provide a clear description of any manipulations performed during this interval and justify their duration.

Response: The interval between egg preparation and tumor formation is necessary for normal embryo development and CAM formation. Specifically, the eggs were incubated for seven days before inoculation, allowing the embryo and CAM to reach a stage at which the membrane is sufficiently vascularized and developed. No manipulations were performed during this interval. The only procedures performed were standard incubation, opening the eggshell on EDD3, and daily inspection of the eggs. A sample of glioblastoma tissue was obtained on day EDD7 of embryo development, and a cell suspension (prepared from the tumor content for seeding on the CAM) was prepared for simultaneous seeding in all cases.

 Line 206. For transparency and reproducibility, please report the exact concentrations used in the mixture of test substances applied to the model.

Response: The concentration of NaVPA in the 20 µL mixture of a single tumor was
2 mM, and that of NaDCA was 3 mM.

 Line 207. Please state whether cell viability was assessed after preparation of the suspension and, if so, which assay was used and the quantitative results. Additionally, estimate how many cells were present in the 20 µL of suspension applied to the gelatin sponge, and describe how that estimate was obtained.

Response: We did not determine cell viability. Tumor growth on CAM, along with histological and histochemical GFPA expression studies, indicates that tumor cells remained viable. Other authors indicate that the processing of patient tumor tissue should not exceed 2–3 hours (6–8).

We did not evaluate cell viability because the entire sample was used immediately after processing, as the aim of the study was to preserve the complete heterogeneity of the primary cells and avoid the loss of fragile primary cell properties during additional processing steps. In addition, to avoid loss of viability, the cells were implanted immediately after the implantable tumor content formed.

We did not count the cells because the suspension contained not only individual cells but also very small pieces of tumor tissue. Our experience shows that tumor studies can also be performed by transplanting tumor tissue onto CAM (9–12). Our measure was the uniform-suspension-volume transfer on CAM.

References

9.     Uloza V, Kuzminienė A, Palubinskienė J, Balnytė I, Ulozienė I, Valančiūtė A. Laryngeal carcinoma experimental model suggests the possibility of tumor seeding to gastrostomy site. Med Hypotheses. 2021 May;150:110573. doi: 10.1016/j.mehy.2021.110573.

10.  Uloza V, Kuzminienė A, Palubinskienė J, Balnytė I, Ulozienė I, Valančiūtė A. An Experimental Model of Human Recurrent Respiratory Papillomatosis: A Bridge to Clinical Insights. Laryngoscope. 2021 Mar;131(3):E914-E920. doi: 10.1002/lary.29093.

11.  Uloza V, Kuzminienė A, Šalomskaitė-Davalgienė S, Palubinskienė J, Balnytė I, Ulozienė I, Šaferis V, Valančiūtė A. Effect of Laryngeal Squamous Cell Carcinoma Tissue Implantation on the Chick Embryo Chorioallantoic Membrane: Morphometric Measurements and Vascularity. Biomed Res Int. 2015;2015:629754. doi: 10.1155/2015/629754. 4.

12.  Balciūniene N, Tamasauskas A, Valanciūte A, Deltuva V, Vaitiekaitis G, Gudinaviciene I, Weis J, von Keyserlingk DG. Histology of human glioblastoma transplanted on chicken chorioallantoic membrane. Medicina (Kaunas). 2009;45(2):123-31.

Line 212. Please report the number of embryonated eggs used per experimental group (per sample), and indicate whether this number was determined by a power calculation or other rationale.

Response: Table 1 indicates that n also represents the number of eggs. For GBM2-2F patient tissue samples, 8 eggs were used in the control group and 8 eggs in the NaVPA–NaDCA treatment group. For GBM2-3F, 10 eggs were used in each group. In the case of GBM2-4M, 7 eggs were used in the control group and 7 eggs in the treatment group. The number of studies was based on established CAM research practice, which shows that a group of 7–12 eggs is sufficient. Typical embryo mortality is 10–20% before the formed tumor is implanted. The formed tumor was implanted on viable embryos, and the studies were completed successfully in all groups after tumor transplantation.

Line 213. Was the stated incubation period sufficient for tumour formation on the CAM? Please provide evidence (references or pilot data) supporting the chosen incubation duration for reliable tumour establishment.

Response: The incubation period used in our study (from EDD7 to EDD12; 5 days) is sufficient for CAM to form a reliable tumor. This is confirmed by our previous studies and the literature sources listed below:

Skredėnienė R, Stakišaitis D, Valančiūtė A, Balnytė I. In Vivo and In Vitro Experimental Study Comparing the Effect of a Combination of Sodium Dichloroacetate and Valproic Acid with That of Temozolomide on Adult Glioblastoma. Int J Mol Sci. 2025 Jul 15;26(14):6784. doi: 10.3390/ijms26146784.

Stakišaitis D, Damanskienė E, Curkūnavičiūtė R, Juknevičienė M, Alonso MM, Valančiūtė A, Ročka S, Balnytė I. The Effectiveness of Dichloroacetate on Human Glioblastoma Xenograft Growth Depends on Na+ and Mg2+ Cations. Dose Response. 2021 Feb 27;19(1):1559325821990166. doi: 10.1177/1559325821990166.

Kavaliauskaitė D, Stakišaitis D, Martinkutė J, Šlekienė L, Kazlauskas A, Balnytė I, Lesauskaitė V, Valančiūtė A. The Effect of Sodium Valproate on the Glioblastoma U87 Cell Line Tumor Development on the Chicken Embryo Chorioallantoic Membrane and on EZH2 and p53 Expression. Biomed Res Int. 2017;2017:6326053. doi: 10.1155/2017/6326053.

Šlekienė L, Stakišaitis D, Balnytė I, Valančiūtė A. Sodium Valproate Inhibits Small Cell Lung Cancer Tumor Growth on the Chicken Embryo Chorioallantoic Membrane and Reduces the p53 and EZH2 Expression. Dose Response. 2018 Apr 26;16(2):1559325818772486. doi: 10.1177/1559325818772486.

Kim SS, Pirollo KF, Chang EH. Isolation and Culturing of Glioma Cancer Stem Cells. Curr Protoc Cell Biol. 2015 Jun 1;67:23.10.1-23.10.10. doi: 10.1002/0471143030.cb2310s67.

Line 217. Please indicate the gauge/size of the needle used for FITC inoculation, and provide a brief justification for this choice.

Response: An insulin syringe was used to inject dextran into the CAM blood vessel because its very thin needle allows precise injection into the small, sensitive vessel, thereby reducing tissue damage and the risk of bleeding. The needle used in the study was manufactured by Chirana T. Injecta in Star Tura, Slovakia, and is 29G in size. We have included the manufacturer's information in Section 2.4 L226.

Line 182. The protocol description is unclear regarding how the mixture of test substances was inoculated onto the CAM. Please provide a step-by-step account of how the tested mixture was prepared and delivered to the model (timing relative to cell inoculation, vehicle used, physical placement on the CAM, etc.).

Response: The GBM cell suspension was centrifuged at 800 rpm for 3 min and the supernatant was discarded. The cells in DMEM were resuspended in rat tail collagen, type I (group – control), and in a mixture of collagen, 2 mM NaVPA and 3 mM NaDCA in DMEM (treated group), total volume of preparation for single tumor formation –
20 µL. A 20 µL of GBM cell suspension mixture was pipetted onto a 4 mm³ piece (1 × 2 × 2 mm) of a hemostatic, absorbable gelatin sponge. At the EDD7, the tumor-bearing sponge was transplanted on the CAM in proximity to major blood vessels. We have corrected the protocol description in Section 2.4 L210-L214.

Line 335. Please describe in detail the methods used to obtain the histomorphometric data (e.g., staining procedures, image acquisition, software, thresholds, number of fields per sample, operator blinding). These methods should appear in the Materials and Methods section, as the current description is insufficient for reproducibility.

Response: We believe that the current description for reproducibility is sufficient. The staining procedures, image acquisition parameters, software used, and number of fields per sample are all described in Section 2.5. However, we have added a sentence describing operator blinding in Section 2.5 L280-L281 to further clarify the methodology. The sentence: Histological and histomorphometric analyses were performed by an operator blinded to the experimental groups.

Figures. Please add explanatory legends beneath each figure that include the sample size (n), summary statistics shown, statistical tests used, exact p-values (or ranges), and definition of error bars. This information should also be referenced in the main text when interpreting the figures.

Response: The sample sizes are reported in Table 2 and Tables A1–A5, and statistics are described in Section 2.6. Exact p-values (or ranges) are provided directly within the Figures. The definition of the error bars is also specified in Section 2.6 (L288-L289), where CAM sub-tumoral thickness and IHC marker expression are described as medians with corresponding minimum and maximum values. We think that including these details in each Figure legend would be redundant and may unnecessarily reduce the manuscript's readability. Therefore, we hope that our description is appropriate.

Line 361. Please specify the time interval after inoculation of the cell suspension at which neovascularization was first observed and how it was assessed (macroscopically, microscopically, or via imaging).

Response: Neovascularization was first observed 2 days after inoculation (at EDD9) using a stereomicroscope at 5× magnification (scale – 1 mm).

The quality of the histological images should be improved for publication. Please provide higher-resolution micrographs, indicate magnification and scale bars, and revise the image descriptions to reflect these changes.

Response: Thank you for a remark. We have improved the quality of the histological images. Magnifications and scale are indicated at the image descriptions. Corrections to the Figures were made based on another reviewer's comment.

Please ensure consistent and correct formatting of the terms in vivo and in vitro throughout the manuscript (italicized).

Response: Corrected according to the comment.

Comments 3:

To strengthen the rigor and reproducibility of the study, several methodological improvements are recommended. First, a dose–response experiment should be conducted using multiple concentrations, reporting IC₅₀/EC₅₀ values where applicable, and increasing the number of biological replicates to enhance statistical power. Second, cell viability and accurate cell counts should be reported. For histomorphometric analyses, full methodological details should be provided, including microscope model and objective magnification, image acquisition and analysis software, analysis parameters, number of sections and fields evaluated per sample, and checks for inter-observer variability.

Response: Thank you for the comment.

Our explanations regarding tumor cell viability, IC₅₀/EC₅₀ values, and cell count are provided above. Concerning the histomorphometric analyses, we confirm that full methodological details, including microscope specifications, image acquisition parameters, software used during the study and analysis parameters are provided in Sections 2.4 and 2.5 of the manuscript. We hope these clarifications adequately address the reviewer’s concerns.

Comments 4:

The limitations of the study are discussed and refer only to the small number of samples used to evaluate the inhibitory potential of the NaVPA–NaDCA combination. My question is whether the results obtained in the CAM model can be extrapolated to animal models with xenograft transplants?

Response: Thank you for your important question. We agree that extrapolation of CAM model results to animal xenotransplantation models should be evaluated in light of the advantages and disadvantages of these models. The CAM assay is widely used as an intermediate in vivo platform for rapid assessment of tumor growth, angiogenesis, and treatment effects, while reducing complexity and ethical constraints relative to rodent models. The CAM model is immunodeficient for up to 12 weeks, does not fully reflect the immune system, and cannot evaluate the long-term interactions between the tumor and the host that are characteristic of mammalian xenotransplantation models. However, as we pointed out in the article, rodent models are largely unsuccessful and their studies are associated with information loss. Therefore, our results provide preliminary evidence that the CAM model is valuable for evaluating the efficacy of the drug under investigation and that results are obtained relatively quickly. Therefore, the CAM model may be important for evaluating drug efficacy before prescribing chemotherapy.

Comments 5:

I consider it important to reassess the conclusions, highlighting the novel aspects revealed by this study. Additionally, any limitations and implications for future research should be clearly addressed to contextualize the findings.

Response: In the discussion section, we added information on the study's novelty and revised the conclusions in accordance with the comment. One new reference (#92) has been added.

4. Response to Comments on the Quality of English Language

Point 1:

Response 1: We have reviewed and corrected the English language. Another reviewer indicated that the English language is good. Therefore, we expect that only minor language corrections may be necessary (in red).

5. Additional clarifications

We thank the Reviewer for important comments. Based on these comments, we have revised the manuscript and provided detailed answers to the questions.

We are grateful for the comments, which have enabled us to improve the manuscript quality. We hope that the revised manuscript version is acceptable for publication.

Sincerely,

Donatas Stakišaitis

Round 2

Reviewer 2 Report

Comments and Suggestions for Authors

The authors have adequately addressed the comments by providing the necessary improvements and clarifications. Consequently, I consider that, in its current form, the manuscript meets the standards of the journal and is suitable for consideration in the subsequent stages of the publication process.